# Hypernetworks in Meta-Reinforcement Learning

**Jacob Beck**
Department of Computer Science
University of Oxford, United Kingdom
`jacob_beck@alumni.brown.edu`

**Matthew Jackson**
Department of Engineering Science
University of Oxford, United Kingdom
`jackson@robots.ox.ac.uk`

**Risto Vuorio**
Department of Computer Science
University of Oxford, United Kingdom
`risto.vuorio@keble.ox.ac.uk`

**Shimon Whiteson**
Department of Computer Science
University of Oxford, United Kingdom
`shimon.whiteson@cs.ox.ac.uk`

**Abstract:** Training a reinforcement learning (RL) agent on a real-world robotics task remains generally impractical due to sample inefficiency. Multi-task RL and meta-RL aim to improve sample efficiency by generalizing over a distribution of related tasks. However, doing so is difficult in practice: In multi-task RL, state of the art methods often fail to outperform a degenerate solution that simply learns each task separately. Hypernetworks are a promising path forward since they replicate the separate policies of the degenerate solution while also allowing for generalization across tasks, and are applicable to meta-RL. However, evidence from supervised learning suggests hypernetwork performance is highly sensitive to the initialization. In this paper, we 1) show that hypernetwork initialization is also a critical factor in meta-RL, and that naive initializations yield poor performance; 2) propose a novel hypernetwork initialization scheme that matches or exceeds the performance of a state-of-the-art approach proposed for supervised settings, as well as being simpler and more general; and 3) use this method to show that hypernetworks can improve performance in meta-RL by evaluating on multiple simulated robotics benchmarks.

**Keywords:** Meta-Learning, Reinforcement, Hypernetwork

## 1 Introduction

Deep reinforcement learning (RL) has helped solve previously intractable problems but still remains highly sample inefficient. This sample inefficiency makes it impractical, particularly in settings where data collection happens in the real world. For example, a robot's actions have the potential to inflict damage on both itself and its surroundings. Multi-task RL and meta-RL aim to improve sample efficiency on novel tasks by generalizing over a distribution of related tasks. However, such generalization has proven difficult in practice. In fact, multi-task RL methods often fail to outperform a degenerate solution that simply trains a separate policy for each task [1].

One promising way to improve generalization is with a *hypernetwork*, a neural network that produces the parameters for another network, called the *base network* [2]. In multi-task RL, using a hypernetwork that conditions on the task ID to generate task-specific parameters can replicate the separate policies of the degenerate solution, while also allowing generalization across tasks. Furthermore, unlike the degenerate solution, hypernetworks can also be applied to meta-RL, where task IDs are not provided and test tasks may be novel, by conditioning them on the output of a task encoder.

However, hypernetworks come with their own challenges. Since hypernetworks generate base network parameters, the initialization of parameters in the hypernetwork determines the initialization of the base network it produces. Evidence suggests hypernetwork performance is highly sensitive to the initialization scheme in supervised learning [3]. However, to our knowledge this question has

6th Conference on Robot Learning (CoRL 2022), Auckland, New Zealand.

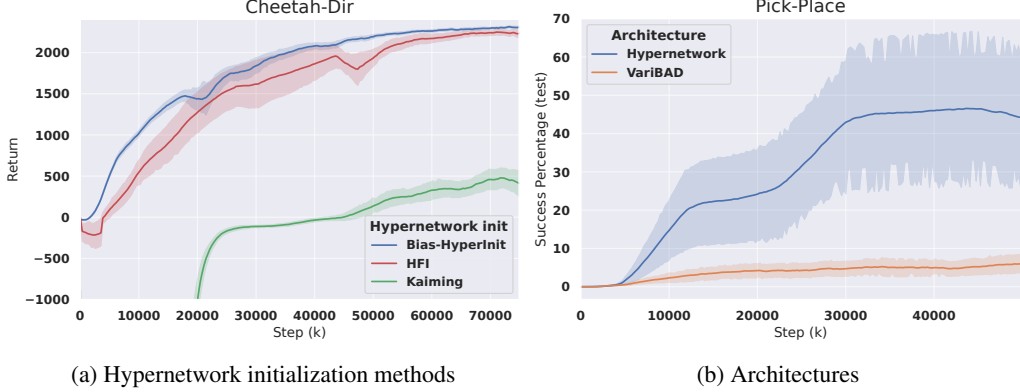

(a) Hypernetwork initialization methods        (b) Architectures

Figure 1: Naive initializations such as Kaiming [4] fail for hypernetworks, whereas our proposed Bias-HyperInit does not and matches the state of the art, HFI [3] (claims 1, 2). Adding hypernetworks with the proposed Bias-HyperInit significantly improves the state-of-the-art meta-RL method, VariBAD [5] (claim 3).

not been considered in meta-RL. In this paper, we show that hypernetwork initialization is also a critical factor in meta-RL, and that naive initializations yield poor performance.

Furthermore, we propose two novel initialization schemes: *Bias-HyperInit* and *Weight-HyperInit*. Both produce strong results, with the former matching or exceeding the performance of the state-of-the-art hypernetwork initialization method designed for supervised learning [3]. Moreover, both proposed methods are simpler and more general than this existing method, in that they may be applied to arbitrary base network architectures and target base network initializations without additional derivation. Using Bias-HyperInit, we present results that substantially improve the a state-of-the-art method on a range of meta-RL benchmarks.

Applying hypernetworks to meta-RL, we make the following contributions (examples in Figure 1):

1. We empirically demonstrate that initialization is a critical factor in the performance of hypernetworks in meta-RL, and that naive initializations fail to learn reliably;

2. We propose a novel hypernetwork initialization scheme that matches or exceeds the performance of a state-of-the-art approach proposed for supervised settings, as well as being simpler and more general; and

3. We use this method to show that hypernetworks can improve a state-of-the-art method on a range of meta-RL benchmarks (grid-world [5], MuJoCo [6], and Meta-World [1]).

## 2 Related Work

**Meta-RL.** Despite the advantages of hypernetworks [2], they remain relatively unexplored in meta-RL. We use hypernetworks to arbitrarily update a policy's parameters at every time-step, whereas all prior work we are aware of restrict this procedure in some way. Many procedures in few-shot meta-RL build off of MAML [7] to adapt the parameters of a policy network using a policy gradient [7, 8, 9]. Such methods require the estimation of a policy gradient, which reduces sample-efficiency when faster adaptation is possible, as in our benchmarks [5]. Most meta-learning procedures capable of zero-shot adaption using an RNN (or convolutions) that can represent an arbitrary update function [5, 10, 11]. These methods generally update a set of activations on which a fixed policy is then conditioned, whereas hypernetworks update all policy parameters. We include a state-of-the-art method from this class in our evaluations [5]. There are also unsupervised methods in zero-shot meta-RL for weight updates [12, 13] but none can produce a fully general learning procedure since they make use of local and unsupervised heuristics. Sarafian et al. [14] use hypernetworks in the context of meta-RL, but the policy network, not the hypernetwork, is conditioned on the RNN used for adaptation, preventing the hypernetwork from representing a general learning procedure. Finally, FLAP [15] learns to infer a set of weights trained in the multi-task setting; however since the adaptation procedure is not trained on a meta-RL objective, it is constrained. For example,

FLAP cannot learn to explore to reduce uncertainty. Finally, Xian et al. [16] use hypernetworks to predict model dynamics then use model predictive control. However, this model still requires planning to make use of an uncertain model, whereas model-free RL learns a policy that explores optimally in order to attain data for adaptation. Using a general procedure trained to arbitrarily modify the weights of a model-free policy has never been tried in RL, to the best of our knowledge.

**Hypernetworks.** Hypernetworks, or similar architectures, have been used in supervised learning (SL), multi-task RL, and meta-SL. Hypernetworks have been used in the supervised learning literature for sequence modelling [2], as well as in continual learning and image classification [3], where it was shown that the hypernetwork initialization scheme was crucial for performance. Similar models have also been used in multi-task RL and meta-SL, but not meta-RL. For instance, in multi-task RL, Yu et al. [17] use a network conditioned on a task encoding to produce the weights and biases for every other layer in another network conditioned on state. In meta-SL, there have also been attempts to use one network to adapt weights of another, both as a general function of the dataset [18, 19, 20], conditioned on an embedding adapted by gradient descent [21], and by adding deltas in a way framed as learning to optimize [22, 23]. The abundance of representations in meta-SL suggest there is a similarly large space of representation-based methods to explore in meta-RL. Our work – getting hypernetworks to work in practice for meta-RL – can be seen as a first step towards applying all of these methods in meta-RL.

## 3 Background

### 3.1 Problem Setting

An RL task is formalized as a Markov Decision Processes (MDP). We define an MDP as a tuple of $(\mathcal{S}, \mathcal{A}, \mathcal{R}, \mathcal{P}, \gamma)$. At time-step $t$, the agent inhabits a state, $s_t \in \mathcal{S}$, observable by the agent. The agent takes an action $a_t \in \mathcal{A}$. The MDP then transitions to state $s_{t+1} \sim \mathcal{P}(s_{t+1}|s_t, a_t) \colon \mathcal{S} \times \mathcal{A} \times \mathcal{S} \to \mathbb{R}_{\geq 0}$, and the agent receives reward $r_t = \mathcal{R}(s_t, a_t) \colon \mathcal{S} \times \mathcal{A} \to \mathbb{R}$ upon entering $s_{t+1}$. Given a discount factor, $\gamma \in [0, 1)$, the agent acts to maximize the expected future discounted reward, $R(\tau) = \sum_{r_t \in \tau} \gamma^t r_t$, where $\tau$ is the agent's trajectory over an episode in the MDP. To maximize this return, the action takes actions sampled from a learned policy, $\pi(a|s) \colon \mathcal{S} \times \mathcal{A} \to \mathbb{R}_+$.

Meta-RL algorithms learn an RL algorithm, i.e., a mapping from the data sampled from a single MDP, $\mathcal{M} \sim p(\mathcal{M})$, to a policy. Since an RL algorithm generally needs multiple episodes of interaction to produce a reasonable policy, the algorithm conditions on $\tau$, which is the entire sequence of states, actions, and rewards within $\mathcal{M}$. As in the RL setting, this sequence up to time-step t forms a trajectory $\tau_t \in (\mathcal{S} \times \mathcal{A} \times \mathbb{R})^t$. Here, however, $\tau$ may span multiple episodes, and so we use the same symbol, but refer to it as a *meta-episode*. The policy is then a meta-episode dependent policy, $\pi_\theta(a|s, \tau)$, parameterized by the *meta-parameters*, $\theta$.

We define the objective in meta-RL as finding meta-parameters $\theta$ that maximize the sum of the returns in the meta-episode across a distribution of tasks (MDPs):

$$\arg\max_\theta \mathbb{E}_{\mathcal{M} \sim p(\mathcal{M})} \left[ \mathbb{E}_\tau \left[ R(\tau) \middle| \pi_\theta(\cdot), \mathcal{M} \right] \right] \tag{1}$$

### 3.2 Policy Architecture

We consider meta-RL agents capable of adaptation at every time-step, and adaptation within one episode is required to solve some of our benchmarks. In such methods [5, 10, 11], the history is generally summarized by a function, $g$, into an embedding that represents relevant task information. We write this embedding as $e = g(\tau)$, and call $g$ the task encoder. The policy, represented as a multi-layer perceptron, then conditions on this task embedding as an input, instead of on the history directly, which we write as $\pi_\theta(a|s, e)$. We call this the *standard architecture*, shown in Figure 2.

In this paper, we primarily build off of VariBAD [5], which can be seen as an instance of the standard architecture where the task encoder is the mean and variance from a recurrent variational auto-encoder (VAE) [24] trained using a self-supervised loss. In other words, the task is inferred as a latent variable optimized for reconstructing a meta-episode. See Zintgraf et al. [5] for details. Additionally, evaluate the addition of hypernetworks to RL2 [11] on the most challenging benchmark. (See section 5.2.) In RL2, the task encoder is a recurrent neural network trained end-to-end on equation 1.

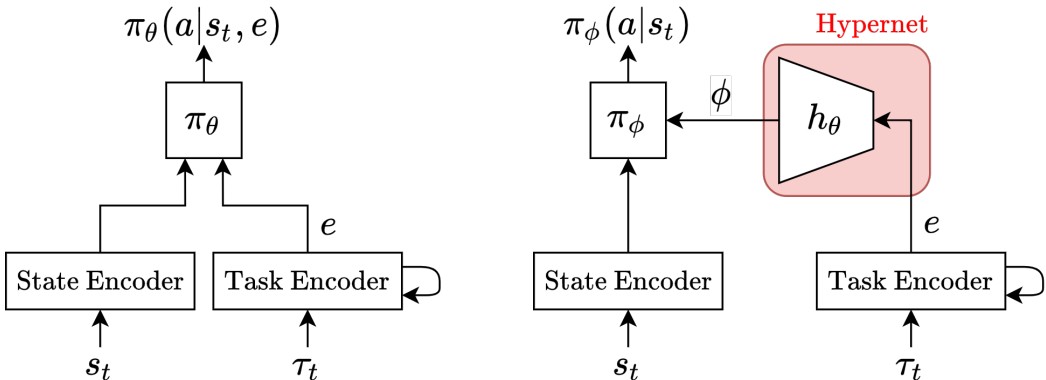

Figure 2: A standard architecture (left) and hypernetwork model (right).

### 3.3 Hypernetwork Initialization

Chang et al. [3] show that applying existing initialization methods for neural networks to hypernetworks produces unstable base network initializations with exploding or vanishing activations. Furthermore, they empirically demonstrate a reduction in training stability for Kaiming initialization [4]. We corroborate this failure for Kaiming initialization on meta-RL, as well as for Orthogonal initialization [25] and Normc initialization [26], which we collectively refer to as *default initializations*.

As a solution, Chang et al. [3] propose the first initialization designed for hypernetworks and show it to be effective in supervised learning. Their approach is based on Kaiming initialization, which samples network parameters such that the activations of the network at each layer maintain the same variance as in the previous layer. Chang et al. [3] extend this variance analysis to hypernetworks [2]. They propose two methods: Hyperfan-in (HFI) sets the variance of the initial parameter distribution of the hypernetwork to maintain constant variance of activations in the base network in the forward pass, and hyperfan-out (HFO) does the same for the backward pass. Since these are equally competitive, and produce state-of-the-art results, we include HFI as a baseline for comparison. However, this variance analysis is involved and requires modification for specific use cases depending on the activation function and whether or not the network produces weights or biases in the base network. This motivates the need for a simpler and more general initialization method, which we propose in this paper.

## 4 Methods

In this section we introduce our proposed architecture using hypernetworks and our proposed hypernetwork initialization. At a high level, the hypernetwork conditions on a task representation to generate all of the parameters for a base policy. The initialization provides a simple and general way to sample parameters for this hypernetwork at the start of training.

### 4.1 Policy Architecture

We propose the use of hypernetworks in meta-RL, instead of the standard architecture described earlier. In this setting, we use a hypernetwork to arbitrarily adapt the parameters of the base policy. We still use a task encoder, $e = g(\tau)$, but instead of conditioning a policy on these activations, we use a hypernetwork, $h_\theta$, to generate policy parameters, $\phi = h_\theta(e)$. These parameters, i.e., weights and biases, are then used directly for the base network, which we write: $\pi_\phi(a|s)$. This is also depicted in Figure 2.

In this case, the hypernetwork can arbitrarily adapt all the the parameters of $\pi$ based on history. In comparison, in the standard architecture, the shared fixed parameters of the base network ($\theta$) are still required to generalize between all of the tasks. Since training separate policies for each task often performs better than state-of-the-art methods for generalizing across all tasks [1], this motivates the ability to produce base policies with no or few shared parameters. Hypernetworks allow for shared

parameters when possible, but also provide the ability to have no shared parameters in the base policy when diverse policies are necessary and little generalization between tasks is possible.

In fact, hypernetworks are capable of replicating the training of separate policies for each task under certain conditions. To see this, consider the case where the hypernetwork is linear and has no bias. Then, the hypernetwork consists only of a weight matrix, $W$. (That is, $\theta = W$.) If this hypernetwork conditions on a one-hot task ID for task i: $e = \mathbb{1}_i$, then the parameters selected by this hypernetwork, $\phi$ are a separate set of parameters for each task: $\phi^i = h(e) = W\mathbb{1}_i$. In other words, training individual networks for each task is equivalent to training a hypernetwork when that hypernetwork is: 1) linear, 2) has no a bias, and 3) is conditioned on a one-hot task ID.

However, we can relax these assumptions and still retain the ability to produce distinct parameters, while also enabling generalization. If we add a bias, we reintroduce shared parameters in the hypernetwork, but they can still produce separate base networks for each task when little transfer between the policies is required. If we relax the one-hot assumption, the network is still capable of producing a one-hot encoding when the tasks are discrete and the task embedding is sufficiently large. Relaxing these restrictions allows for both generalization and the application of hypernetworks to meta-RL.

## 4.2 Hypernetwork Initialization

Default initialization methods fail for hypernetworks. However, given that hypernetworks generalize training separate networks for each task, it must be possible to initialize them as if each of the corresponding base networks were initialized independently, from some given initialization function, $f$, known to train reliably. Using this insight, we propose and evaluate two initialization schemes for the hypernetwork. We propose one where (under some assumptions) each base network is independently initialized from $f$. We also propose one where (under no assumptions) all base networks share an initialization sampled from $f$.

Our first method is *Weight-HyperInit*. Let $W$ and $b$ be the weights and bias in the last layer of our hypernetwork, $h(e)$, respectively. (These parameters are both contained in $\theta$.) We define this weight-only initialization as follows:

$$W_{:,i} := \phi^i \sim f(\phi) \; \forall i, \qquad b := \mathbf{0},$$

where $f$ is an arbitrary initialization scheme for the base network specifying a distribution over a vector of parameters, $\phi$, and $W_{:,i}$ is the $i$-th column of $W$.

Weight-HyperInit reproduces any given initialization for each base network ($\pi_\phi$), under the assumptions that $e = \mathbb{1}_i$ and the hypernetwork is linear. In this case, each column of $W$ is simply a sample from the base scheme, one of which is selected for each task via the one-hot encoding. For example, given the task embedding $e = \mathbb{1}_3$, the following base network initialization $\phi_{\text{init}}$ is produced:

$$\phi_{\text{init}} = We + b = \begin{pmatrix} \phi_1^1 & \phi_1^2 & \phi_1^3 & \cdots \\ \phi_2^1 & \phi_2^2 & \phi_2^3 & \cdots \\ \phi_3^1 & \phi_3^2 & \phi_3^3 & \cdots \\ \vdots & \vdots & \vdots & \ddots \end{pmatrix} \begin{pmatrix} 0 \\ 0 \\ 1 \\ 0 \\ \vdots \end{pmatrix} + \mathbf{0} = \phi^3. \tag{2}$$

Moreover, in the case that there is also no bias, it is also equivalent to training separate networks for each task. Although these assumptions are not met for meta-RL, and so do not hold for our experiments, we find this is still an improvement over default neural network initialization schemes.

Additionally, we propose *Bias-HyperInit*. This bias-only initialization is defined as follows:

$$W_{i,j} := 0 \; \forall i,j, \qquad b := \phi_{\text{shared}} \sim f(\phi).$$

Bias-HyperInit achieves an arbitrary initialization for the base network without any assumptions, by setting the parameters for any task to be the same at initialization. This encourages parameter sharing between base networks at the beginning of training, where possible, but also allows for separate base network parameters to be learned, if necessary. Under any set of assumptions, the

base network is initialised to the following:

$$\phi_{\text{init}} = Wx + b = \begin{pmatrix} 0 & 0 & 0 & \dots \\ 0 & 0 & 0 & \dots \\ 0 & 0 & 0 & \dots \\ \vdots & \vdots & \vdots & \ddots \end{pmatrix} \begin{pmatrix} x_1 \\ x_2 \\ x_3 \\ x_4 \\ \vdots \end{pmatrix} + \phi_{\text{shared}} = \phi_{\text{shared}}, \tag{3}$$

where $x$ is the final hidden layer of the hypernetwork. ($X = e$ in the case of a linear hypernetwork.)

Both methods initialize only $W$ and $b$, which define the head of the network. All other layers in $h$ may be initialized by any default initialization scheme. All such choices are detailed in supplementary materials.

### 4.3 Baselines

**VariBAD & RL2.** See Sec. 3.2.

**FiLM.** FiLM [27] is a convenient baseline situated between hypernetworks and the standard architecture. In FiLM, the hypernetwork generates biases for each layer, but only point-wise scales the activations instead of generating weight matrices. In this case, the base network has its own weights. Bias-HyperInit can easily be adapted to FiLM; details presented in supplementary materials.

**HFI.** HFI [3] is a state-of-the-art initialization method developed tested in the supervised learning setting. While we do compare to HFI, our methods are simpler and more general. Specifically, our methods work with arbitrary base network target initialization (as opposed to being tied to Kaiming) and our methods work with arbitrary base network architectures (without additional variance analysis). Moreover, our approach is straightforward to apply to additional methods, which we show by applying it to FiLM. Finally, although both HFI and Weight-HyperInit do make assumptions about the input to the hypernetwork, our strongest method, Bias-HyperInit, does not.

## 5 Experiments

In this section, we compare our hypernetwork architecture and initialization methods to baselines on 2D navigation [5], MuJoCo [1], ML1 [1], and ML10 [1] benchmarks. MuJoCo is a common meta-RL benchmark [7, 5, 28, 29, 12], as are toy 2D navigation tasks [7, 5, 28, 29]. These two benchmarks allow us to demonstrate that hypernetworks with default initialization methods fail to learn, whereas our proposed methods learn reliably. ML1 and ML10 are benchmarks in Meta-World [1]. These two benchmarks have greater room for improvement with state-of-the-art methods [5], which allow us to demonstrate improvement over the baseline architectures. Finally, we use two MuJoCo environments to investigate the performance of hypernetworks against standard architectures in terms of the number of parameters in the model overall.

Throughout our evaluation, we use two-tailed $t$-tests with $p = 0.05$ to determine significance. Details on model tuning and implementation are presented in supplementary materials.

### 5.1 Navigation and MuJoCo

Here we evaluate on the grid-world variant from Zintgraf et al. [5] as our 2D navigation task and MuJoCo [6]. Note Grid-World and Cheetah-Dir contain twenty-four and two non-parametric tasks respectively, while the other MuJoCo environments have parametric variation between the tasks.

In Table 1 we see that default initializations frequently fail to learn while Bias-HyperInit learns reliably. Specifically, Kaiming and Normc initializations used with hypernetworks achieve far lower returns than all other methods. Orthogonal initialization is more competitive, however it is still significantly outperformed by Bias-HyperInit in every environment.

We also compare hypernetworks with Bias-HyperInit to the standard architecture and Bias-HyperInit to HFI. We see that Bias-HyperInit matches or exceeds the performance of HFI, with a significant improvement in Walker. Hypernetworks with Bias-HyperInit also significantly outperform the

Table 1: Comparison of return on grid-world and MuJoCo tasks over five random seeds (mean ± standard error). Entries in bold have insignificant difference from the highest-performing result.

| Method | Grid-World | Cheetah-Dir | Walker | Ant-Dir | Humanoid |
|---|---|---|---|---|---|
| Standard | $35.5 \pm 0.4$ | $\mathbf{2104 \pm 87}$ | $\mathbf{1828 \pm 38}$ | $1167 \pm 16$ | $\mathbf{1842 \pm 233}$ |
| Kaiming | $32.1 \pm 1.2$ | $378 \pm 169$ | $331 \pm 37$ | $253 \pm 86$ | $266 \pm 23$ |
| Normc | $32.2 \pm 0.5$ | $356 \pm 134$ | $357 \pm 60$ | $264 \pm 106$ | $249 \pm 18$ |
| Orthogonal | $34.9 \pm 0.5$ | $1379 \pm 310$ | $1687 \pm 98$ | $1127 \pm 93$ | $1126 \pm 76$ |
| HFI | $\mathbf{36.8 \pm 0.2}$ | $\mathbf{2218 \pm 80}$ | $1618 \pm 130$ | $\mathbf{1370 \pm 9}$ | $1323 \pm 57$ |
| Weight-HyperInit | $\mathbf{36.1 \pm 0.5}$ | $\mathbf{2066 \pm 119}$ | $1748 \pm 57$ | $\mathbf{1346 \pm 7}$ | $1048 \pm 21$ |
| Bias-HyperInit | $\mathbf{36.7 \pm 0.2}$ | $\mathbf{2300 \pm 32}$ | $\mathbf{1994 \pm 67}$ | $1328 \pm 23$ | $\mathbf{1678 \pm 162}$ |

standard architecture on grid-world and Ant-Dir. In fact, Bias-HyperInit is not significantly outperformed by any other method. However, the standard architecture, HFI, and Bias-HyperInit all achieve near optimal performance, motivating an evaluation on Meta-World, on which the standard architecture has greater room for improvement [5].

## 5.2 ML1 and ML10

Here we evaluate on the more challenging Meta-World ML1 and ML10 benchmarks [1]. ML1 and ML10 have one and ten non-parametric training tasks respectively (e.g. pushing a ball or opening a window). ML10 additionally has five distinct test tasks. Within each task, there exists parametric variation, such variation in the goal location. Note that we only test on the Pick-Place task from ML1, since VariBAD already achieves a 100% success rate on all other tasks [5].

Table 2: Comparison of meta-test success percentage on the Pick-Place ML1 task (ten seeds) and ML10 (three seeds).

| Method | | Pick-Place | ML10 | |
|---|---|---|---|---|
| | | VariBAD | VariBAD | RL2 |
| Standard | | $4.4 \pm 2.4$[1] | $10.2 \pm 3.0$ | $7.2 \pm 5.0$ |
| FiLM | Normc | $5.5 \pm 4.8$ | — | — |
| | Bias-HyperInit | $\mathbf{34.2 \pm 15.9}$ | — | — |
| Hypernetwork | HFI | $\mathbf{25.5 \pm 14.5}$ | $28.4 \pm 6.0$ | $7.1 \pm 2.4$ |
| | Bias-HyperInit | $\mathbf{42.9 \pm 16.3}$ | $23.9 \pm 6.2$ | $14.2 \pm 7.2$ |

In Table 2 we see significant improvement from hypernetworks with Bias-HyperInit over the standard architecture, as well as the efficacy of Bias-HyperInit on FiLM. On Pick-Place, Bias-HyperInit outperforms the standard architecture with a 9-fold increase in test success percentage. Additionally, Bias-HyperInit improves the FiLM architecture and exceeds the performance of HFI. On ML10, hypernetworks with both Bias-HyperInit and HFI yield a 2-fold increase in test success percentage compared to the standard architecture. Finally, we evaluated Bias-HyperInit when applied to RL2 on ML10, finding a 2-fold increase over both HFI and the standard architecture. Taken together, these results show a clear improvement from the application of hypernetworks with Bias-HyperInit, regardless of the baseline method they are applied to.

---

[1]Zintgraf et al. [5] report a success percentage of 29% for Pick-Place; however, we were not able to replicate this result.

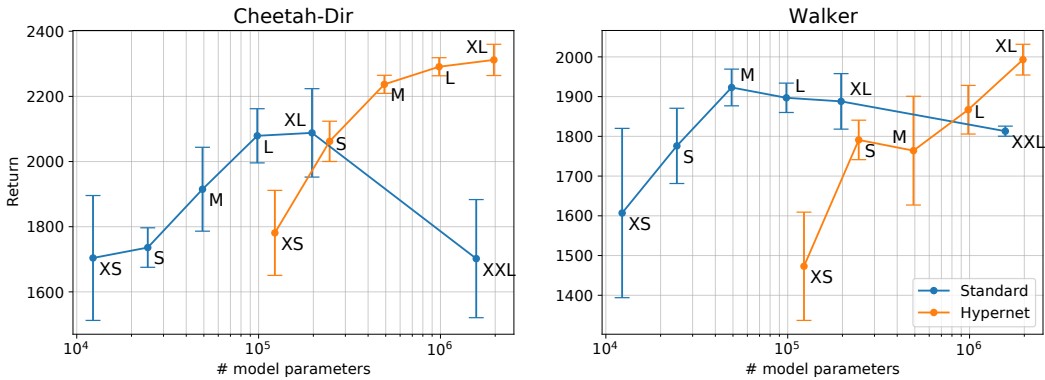

Figure 3: Performance of standard and hypernetwork models over a range of base policy architecture sizes on Cheetah-Dir and Walker. Architectures are presented in supplementary materials.

## 5.3 Network Size Comparison

Because hypernetworks learn a mapping from a task embedding to the parameters of a base network, they require significantly more parameters in the entire model than a standard architecture with the equivalent base network. For a fair comparison, we evaluate return over both a range of base network sizes and total number of parameters in the model on the Cheetah-Dir and Walker tasks (Figure 3). We find that hypernetworks consistently equal or outperform the standard architectures with the equivalent base network size. We also find that hypernetworks likewise outperform standard architectures for equivalent number of parameters in the entire model, i.e. for a given value on the x-axis, when the total number of parameters in the model is sufficiently large.

## 6 Limitations

While our proposed methods are general, we cannot guarantee an improvement for all meta-RL methods. To mitigate this limitation, we build on top of VariBAD, which is state of the art, and additionally evaluate our method applied to RL2 on ML10. Furthermore, as in any empirical study, there is no guarantee that our results hold on real robots or other meta-RL benchmarks. However, we have tested on seven standard meta-RL environments in total, including Meta-World, which was proposed specifically for addressing robotics. As much as is possible from simulated meta-RL experiments, these results give us confidence in a significant improvement over previous methods.

## 7 Conclusion

We used hypernetworks to improve a state-of-the-art method in meta-RL, evaluating over a range of benchmarks. In doing so, we demonstrated that hypernetworks are a promising path forward for meta-RL. Moreover, we showed that the initialization of the hypernetwork is crucial, as default initialization methods fail. To overcome this difficulty, we presented two novel initialization methods: Bias-HyperInit and Weight-HyperInit. Bias-HyperInit matched or exceeded the performance of existing methods from the supervised learning setting, while also being simpler and more general – applying to arbitrary base network initializations, base network architectures, and also improving FiLM. Using Bias-HyperInit, we showed that hypernetwork performance improves substantially over standard architectures. Finally, we demonstrated that hypernetworks outperform the standard architecture for equivalently sized base policies, and outperform it at any size given sufficiently many parameters in the entire model. This paper additionally opens the path for future research extending meta-SL methods using hypernetworks [18, 22] and multi-task RL methods with separate parameters for each task [30, 31, 32] to meta-RL.

**Acknowledgments**

We would like to thank Luisa Zintgraf for her help with the VariBAD code-base along with general advice and discussion. Jacob Beck is supported by the Oxford-Google DeepMind Doctoral Scholarship. Matthew Jackson is supported by the UK EPSRC CDT in Autonomous Intelligent Machines and Systems, with funding from AWS in collaboration with the Oxford-Singapore HMC Initiative. Risto Vuorio is supported by EPSRC Doctoral Training Partnership Scholarship and Department of Computer Science Scholarship.

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
