# OpenReview forum: "Hypernetworks in Meta-Reinforcement Learning"
_robot-learning.org/CoRL/2022/Conference — CoRL 2022 Poster_

### Official Review · Reviewer_7NSX · 2022-07-25

**Originality:** Very Good
**Technical Quality:** Excellent
**Clarity Of Presentation:** Excellent
**Impact:** 3

**Recommendation:**

Weak Accept: I recommend accepting the paper, but will not argue for my recommendation if the majority of other reviewers have a different opinion.

**Summary:**

This paper proposed using hypernetwork in Meta Reinforcement Learning problem. The authors stated that initialization is crucial for the hypernetwork performance. Thus, they proposed and tested two initialization methods: Weight-HyperInit and Bias-HyperInit. In this work, the author run experiment on 2D navigation, MuJoCo, ML1 and ML10 benchmarks in Meta RL setting. Using VariBAD algorithm as baselines, they showed the effectiveness of their method on those baselines. They also compared the performance of using hypernetwork and VariBAD baseline with the equivalent base network size. Also, they pointed out that this method still has some limitations and needs to be tested on real robots.

**Issues:**

1. In section 5.2, the variance of Bias-HyperInit is big. Does that mean using hypernetwork is still sensitive to initialization in Pick-Place tasks?
2. Section 5.3, the author mentioned comparing performance for equivalent number of parameters in the entire model. But I could not find the related results.
3. The proposed hypernetwork architecture is quite straightforward. Does that mean a near-optimal policy for each meta-RL problems in the experiment can be considered of having a linear transform in the policy parameter space?


**Quality Of The Limitations Section:**

Limitations are addressed clearly

**Reviewer Expertise:**

4: The reviewer is confident but not absolutely certain that the evaluation is correct

**Robotics Focus:**

Highly relevant to robotics but no hardware experiments

**Strengths And Weaknesses:**

Strength
1. Well stated motivation in using hypernetwork for meta reinforcement learning.
2. Good performance in control problems.
3. Clear analysis in the scalability of the hyper-network.

Weakness
1. The meta-RL problems are variants between different RL task. Evaluating in meta-RL where each tasks differ in the goal context would make this paper stronger.
2. Specifying how many tasks are used for meta-training stage and meta-test stage will make this paper stronger.


**Summary Of Recommendation:**

I recommend weak-accept of this paper. The proposed method is efficient and the paper is well written. The experiment showed the effectiveness of the proposed method although some of the details might be missed.

---

> ### Author Response · Authors · 2022-08-19
> **Response to Reviewer 3**
>
> Thank you for your feedback. We appreciate you noting that the paper shows good performance, has clear analysis, and is very convincing. We address your concerns below and have incorporated your feedback into the updated version of the paper. Please let us know if any topic remains unclear.
>
> Weaknesses:
> 1. **“Evaluating in meta-RL where each tasks differ in the goal context would make this paper stronger”:** Several of our tasks do already vary by goal location: gridworld, cheetah-dir, ant-dir, ML1, ML10. We will clarify this.
> 2. **“Specifying how many tasks are used for meta-training stage and meta-test stage will make this paper stronger”:** We will add these details.
>
> **Specifically, to address 1 and 2, we have added the following details to section 5:**
> Grid-World and Cheetah-Dir contain twenty-four and two non-parametric tasks respectively, while the other MuJoCo environments have parametric variation between the tasks. ML1 and ML10 have one and ten non-parametric training tasks respectively (e.g. pushing a ball or opening a window). ML10 additionally has five distinct test tasks. Within each task, there also exists parametric variation, such variation in the goal location.
>
> Issues:
> 1. **“In section 5.2, the variance of Bias-HyperInit is big. Does that mean using hypernetwork is still sensitive to initialization in Pick-Place tasks?”:** Many seeds fail to learn on the Meta-World benchmark. This is not uncommon. Our primary benchmark, VariBAD reports that only 6 out of 20 seeds work for Pick-Place, even without hypernetworks (Zintgraf et al., 2021). Since this phenomenon occurs for other methods, it is likely due to the difficulty of the benchmark and not an issue with initialization.
> 2. **“Section 5.3, the author mentioned comparing performance for equivalent number of parameters in the entire model. But I could not find the related results”:** This information is in Figure 3. The x-axis of Figure 3 is the total number of parameters in the entire model. The point here is that, past a certain point on the x-axis, hypernetworks start to outperform the standard architecture (they are higher on the y-axis). To clarify this, we will add “i.e. for a given value on the x-axis”. (The second point in this section is that hypernetworks outperform the standard architecture when comparing equivalent base network sizes, e.g. all points labeled XL denote the XL base-network size.)
> 3. **“Does that mean a near-optimal policy for each meta-RL problems in the experiment can be considered of having a linear transform in the policy parameter space?”:** Not necessarily. The task encoder consists of a recurrent neural network and a feedforward layer. This transformation of the observations is not linear. However, it is the case that most near-optimal policies can be represented as a linear transformation of some non-linear task embedding.

---

### Official Review · Reviewer_R1kq · 2022-07-27

**Originality:** Good
**Technical Quality:** Fair
**Clarity Of Presentation:** Very Good
**Impact:** 2

**Recommendation:**

Weak Reject: I recommend rejecting the paper, but will not argue for my recommendation if the majority of other reviewers have a different opinion.

**Summary:**

The paper has two contributions: 1. it demonstrates the use of hypernetworks for meta reinforcement learning and shows that such approach outperforms VariBAD; 2. it proposes a novel initialization scheme for hypernetworks which makes training possible.

**Issues:**

See the strengths&weaknesses section.

**Quality Of The Limitations Section:**

Additional details required

**Reviewer Expertise:**

4: The reviewer is confident but not absolutely certain that the evaluation is correct

**Robotics Focus:**

Relevant but unlikely to deploy to hardware in near future

**Strengths And Weaknesses:**

The paper is clearly written and easy to follow. Its main downside is that it is unclear whether the authors want to make a contribution to meta-RL or to research on hypernetworks. In its current form, it doesn't present enough experimental evidence to be considered a significant contribution to either of the two fields.

Specifically, given that the main novelty is the proposed initialization scheme for hypernetworks, I would expect to see supervised learning experiments since those are the benchmarks that the baselines were evaluated on (see e.g. the HFI paper).

Regarding meta-RL, I would expect to see comparisons to more baselines than just VariBAD. In particular, you report 19% meta-test success rate for ML10. Looking at the meta-world paper, they report 35.8% meta-test success rate for ML10. You report 5.6% for VariBAD so why is it considered the state-of-the-art?

**Summary Of Recommendation:**

Paper needs more thorough experiments evaluating the proposed methods.

---

> ### Author Response · Authors · 2022-08-19
> **Response to Reviewer 2**
>
> Thank you for feedback. We appreciate you noting that the paper is original, clear, and well-organized. Overall, It seems your main suggestions are to evaluate on supervised benchmarks and include more RL baselines. We are able to directly address the latter concern. While we agree that evaluating the methods on supervised learning would be interesting future work, we believe that our demonstration of their effectiveness in meta-reinforcement learning is still conclusive and impactful to the field of meta-RL.
>
> With respect to the meta-RL experiments, we agree that RL2 would make for a nice addition to the paper and have run additional RL2 experiments. These have been added to the paper. Some comments on this are below:
> 1. **“Why is it considered the state-of-the-art?”:** VariBAD is generally considered a state-of-the-art method (Zintgraf et al., 2021) and commonly used as a stronger baseline than RL2 (Fu et al., 2021, Zhang et al., 2021, Xiong et al., 2021), while RL2 is an older seminal method (Duan et al., 2016). We expect that the 35.8% meta-test success rate for RL2 is due the authors running over a much larger compute budget. To elaborate, in appendix C of their paper, RL2 achieves a meta-test success rate below 20% with a budget of 3.5e8 environment steps. They note that more training was required to achieve the improved results, but we do not see the exact number of environment steps recorded. On the other hand, we run all algorithms for 1e8 steps, which already takes one or more weeks on our hardware.
> 2. **“I would expect to see comparisons to more baselines than just VariBAD”:** To provide further evidence for our methods in meta-RL, we have run RL2 on ML10, as suggested. Results show that VariBAD is indeed a stronger baseline than RL2, as expected. Moreover, results show that the addition of our method likewise improves the RL2 baseline. Additionally, we expanded the range of learning rates on this benchmark, which has improved success percentages a fair amount. Taken together, we believe that our experiments demonstrate a clear improvement from the application of our method, given an equal amount of compute for each method, regardless of the baseline we choose to build on top of.

---

### Official Review · Reviewer_jmaN · 2022-08-03

**Originality:** Fair
**Technical Quality:** Fair
**Clarity Of Presentation:** Fair
**Impact:** 2

**Recommendation:**

Weak Reject: I recommend rejecting the paper, but will not argue for my recommendation if the majority of other reviewers have a different opinion.

**Summary:**

Meta-RL is an important direction to apply RL to real-world applications which require an ability to adapt to new tasks/environments. This paper tackles meta-RL using hypernetworks that produces the parameters of main agent (i.e., another neural networks). First, the authors show that performance can be very sensitive to initialization of hypernetworks and propose two initialization schemes. The first one is not sharing parameters between tasks (i.e., no bias) and independently initializing parameter per each task and the second one is only utilizing the shared one. The authors evaluate the proposed methods on several meta-rl benchmarks.

**Issues:**

* As mentioned in Weaknesses, it would be nice if the authors could clarify the confusing parts (e.g., structure of hypernetworks, problem formulation and so on).


**Quality Of The Limitations Section:**

Limitations are addressed clearly

**Reviewer Expertise:**

4: The reviewer is confident but not absolutely certain that the evaluation is correct

**Robotics Focus:**

Relevant but unlikely to deploy to hardware in near future

**Strengths And Weaknesses:**

* Strengths

1. Interesting research direction: the idea of utilizing hypernetwork for meta-RL sounds interesting (but I also expect that it also has some limitations; see `Weaknesses`)

* Weaknesses

1. writing: I may miss something but the learning components for hyper-network are not clear. At the beginning, I thought that this framework is training NNs (hypernetworks parameterized by $\theta$) which generates parameters ($\phi$) of main agents. However, from Section 4.2, $\theta$ is gone and parameters of hypernetwork ($W,b$) become the parameters of main agent ($\phi$). Overall, current description is bit confusing and hard to understand.

2. soundness: several components are not convincing. First, the authors mentioned that task id is not given in meta-rl but they assume that task id (one-hot vector) is given as inputs to hypernetworks. Also, the proposed initializations just look like independent initialization or one shared parameters. Because of that, it is bit hard to say that the proposed method is convincing and novel.

**Summary Of Recommendation:**

I'd like to recommend "weak reject" due to the weaknesses mentioned above.

---

> ### Author Response · Authors · 2022-08-19
> **Response to Reviewer 1**
>
> Thank you for feedback. We appreciate you noting that applying hypernetworks to meta-RL is an interesting research direction. We address your concerns below and have incorporated your feedback into the updated version of the paper. Please let us know if any topic remains unclear:
> 1. **“From Section 4.2, θ is gone and parameters of hypernetwork (W,b) become the parameters of main agent (ϕ)”:** As you note, θ parameterizes the hypernetwork. In section 4.2. we discuss the weights and bias of the last linear layer, W and b (contained in θ). As noted at the end of section 4.2, our proposed initialization methods “initialize only W and b, which define the head of the network. All other layers in h may be initialized by any default initialization scheme.” To clarify this and address your concern, we have additionally added that “These parameters are both contained in θ” to section 4.2 and “That is, θ = W” in section 4.1. Also, we have now added to the top of section 4.1 that “In the reported experiments, we use a one-layer linear hypernetwork. In principle, our hypernetworks may use any architecture with a linear final layer.” We hope these collectively clarify any confusion about the hypernetwork architecture.
> 2. **A) “First, the authors mentioned that task id is not given in meta-rl but they assume that task id (one-hot vector) is given as inputs to hypernetworks”:** The motivation for Weight-HyperInit assumes that the task ID is one-hot. However, our experiments demonstrate that, in practice, this assumption does not need to hold for Weight-HyperInit to be an effective method. Rather than modifying our experiments to fit this assumption, we address the more difficult problem. That is, we opt to break this assumption in our experiments, and then we verify that Weight-HyperInit is robust to this violation. Even in violating this assumption in our experiments, “we find that this is still an improvement over default neural network initialization schemes.” Reasoning for why this may be the case can be seen at the end of 4.1. To clarify that this assumption is for the motivating analogy only, we have now added an explanation that these assumptions are not met for meta-RL “and so do not hold for our experiments” to 4.2. Equally importantly, Bias-HyperInit does not make any assumption. As we note, “Bias-HyperInit achieves an arbitrary initialization for the base network without any assumptions.” **B) “The proposed initializations just look like independent initialization or one shared parameters. Because of that, it is bit hard to say that the proposed method is convincing and novel”:** It is true that the initialization method sets the head of the hypernetwork such that it produces separate networks for each task (under assumptions that do not hold in our experiments) in Weight-HyperInit, or a shared network for all tasks (i.e. ignore the task completely) in Bias-HyperInit. However, both of these are only initializations of the hypernetwork: the hypernetwork will still learn to arbitrarily modify the base network. In fact, the hypernetwork must learn to take the task into account in Bias-HyperInit, despite its initialization, in order for the base policy to vary between tasks. We believe that both methods constitute novel insights. For instance, it is not obvious that Weight-HyperInit is still useful when one-hot task IDs are not possible, as in many of our meta-RL tasks. It is also not obvious that ignoring the task at initialization, as in Bias-HyperInit, is effective when tasks need to be taken into account. Our experiments demonstrate that these methods, in addition to being simple, also vastly outperform generic neural network initialization methods and match complex hypernetwork initialization methods, making them a novel and impactful contribution.

---

### Meta-Review · Area_Chair_bdyb · 2022-08-08

**Recommendation:** Accept (Poster)
**Confidence:** 4

**Metareview:**

Strengths:
- Novel research direction
- Good empirical results in simulation for some tasks

Weaknesses:
- Some important parts of the method are not clear (e.g., how is task id inferred by the network)
- Only baseline is VariBAD, which is not SOTA on some of the tasks.
- No comparison of the proposed initialization in supervised learning

Post rebuttal/discussion:

The authors clarified the unclear points raised in the reviews, and added a comparison with RL^2.
While supervised learning experiments will strengthen this paper, there is sufficient interest here to CoRL audience even without them.
Finally, all reviewers agreed that this paper can be accepted.

---

> ### Author Response · Authors · 2022-08-19
> **Response to Meta Reviewer**
>
> Thank you for your feedback. We appreciate you noting the novel research direction and good empirical results. Please see responses below:
>
> 1. **“Some important parts of the method are not clear (e.g., how is task id inferred by the network)”:** There were many suggestions in the reviews surrounding clarity. We believe we have addressed these. For a detailed response, please see the individual reviews. Additionally, the meta-review comments that it is unclear how the task ID is inferred. In section 3.2 we note that “the task encoder is the mean and variance from a recurrent variational auto-encoder (VAE) trained using a self-supervised loss,” and we cite VariBAD. To further clarify, we will add that “in other words, the task is inferred as a latent variable optimized for reconstructing a meta-episode.” Please see VariBAD (Zintgraf et al., 2021) for more details.
> 2. **“Only baseline is VariBAD, which is not SOTA on some of the tasks”:** With our compute budget, we were able to add RL2, as suggested, on the ML10 benchmark. Results show that VariBAD is indeed a stronger baseline than RL2, as expected. Moreover, results show that the addition of our method likewise improves the RL2 baseline. Please see review 2 for more details.
> 3. **“No comparison of the proposed initialization in supervised learning”:**  While we agree that evaluating the methods on supervised learning would be interesting future work, we believe that our demonstration of their effectiveness in meta-reinforcement learning is still conclusive and impactful to the field of meta-RL.
>
> Additionally, we summarize our responses by review. We believe we have resolved nearly all of the concerns and adjusted the paper accordingly. We summarize here:
>
> Reviewer 1 raised concerns about identifying parameters, making assumptions, and novelty. We believe all of these were instances of miscommunication. We have addressed these issues in our revision of the paper. For detailed response please see the comment to the review.
>
> Reviewer 2 suggested evaluating supervised benchmarks and including more RL baselines. We were able to directly resolve the concern about RL baselines. Please see the response above for a summary, or see the comment to the review for more details.
>
> Reviewer 3 raised concerns about varying the goal location and the number of tasks in each experiment, in addition to a few other issues of clarity. We believe all of these were instances of miscommunication. We have addressed these issues in our revision of the paper. For detailed response please see the comment to the review.
>
> Please let us know if any topic remains unclear or further adjustments need to be made.